# Faster proximal algorithms for matrix optimization using Jacobi-based eigenvalue methods

**Hamza Fawzi**       **Harry Goulbourne**
Department of Applied Mathematics and Theoretical Physics
University of Cambridge, UK
`{hf323, hmg42}@cam.ac.uk`

## Abstract

We consider proximal splitting algorithms for convex optimization problems over matrices. A significant computational bottleneck in many of these algorithms is the need to compute a full eigenvalue or singular value decomposition at each iteration for the evaluation of a proximal operator.

In this paper we propose to use an old and surprisingly simple method due to Jacobi to compute these eigenvalue and singular value decompositions, and we demonstrate that it can lead to substantial gains in terms of computation time compared to standard approaches. We rely on three essential properties of this method: (a) its ability to exploit an approximate decomposition as an initial point, which in the case of iterative optimization algorithms can be obtained from the previous iterate; (b) its parallel nature which makes it a great fit for hardware accelerators such as GPUs, now common in machine learning, and (c) its simple termination criterion which allows us to trade-off accuracy with computation time. We demonstrate the efficacy of this approach on a variety of algorithms and problems, and show that, on a GPU, we can obtain 5 to 10x speed-ups in the evaluation of proximal operators compared to standard CPU or GPU linear algebra routines. Our findings are supported by new theoretical results providing guarantees on the approximation quality of proximal operators obtained using approximate eigenvalue or singular value decompositions.

## 1   Introduction

Many problems in statistics and learning can be formulated as convex optimization over matrices. This includes kernel learning [LCB$^+$04], low-rank matrix recovery [RFP10, CR09], sparse covariance selection [BGdN06, FHT08], community detection [BBV16], and many others. Proximal splitting algorithms such as (accelerated) proximal gradient methods [Nes03, BT09], or the alternating direction method of multipliers (ADMM) [EB92, BPC11], are among the most popular methods to deal with large-scale instances of such problems. These methods solve generic convex optimization problems of the form

$$\min_{X \in E} \ F(X) + G(X) \tag{1}$$

via calls to the *proximal operators* of $F$ and/or $G$. When the variable $X$ is a symmetric or rectangular matrix, it is often the case that the functions $F$ and/or $G$ are symmetric functions of the eigenvalues or singular values of $X$; i.e., so-called *spectral or othogonally-invariant functions*. Typical examples are $F(X) = I(X \succeq 0) = \sum_{i=1}^{n} I(\lambda_i(X) \geq 0)$ the convex indicator function for the positive semidefinite cone, or $F(X) = \|X\|_* = \sum_{i=1}^{n} \sigma_i(X)$ the nuclear norm.

Computing the proximal operator of such spectral functions $F$ requires a full eigenvalue/singular value decomposition which contributes a large computational cost of at least $n^3$ floating-point

35th Conference on Neural Information Processing Systems (NeurIPS 2021).

operations per iteration. The most common algorithms for such decompositions are the QR method, and the divide-and-conquer algorithm [GVL13] which are implemented in linear algebra packages (LAPACK) that are used in standard numerical computing software such as MATLAB or NumPy.

**Contributions**   In this paper we propose to use a simple and lesser known method due to Jacobi from 1846 [Jac46] for the computation of these decompositions. Several features of this method make it suitable for its use within iterative proximal methods:

(i) It can effectively exploit an initial approximate decomposition of the target matrix. In iterative proximal methods, the eigenvectors/singular vectors from the previous iterate are often a very good initial point that almost diagonalize the matrix of the current iterate.

(ii) The method can be very easily parallelized, and is a very good fit for parallel hardware accelerators, notably GPUs, which are now very common in machine learning.

(iii) It has a simple termination criterion that allows us to trade-off computation time with the accuracy of the desired decomposition.

The present paper has two main contributions. First, we establish new theoretical results providing guarantees on the approximation quality of the proximal operator of a convex spectral function when computed with an approximate eigenvalue/singular value decomposition. When combined with existing results on the convergence of proximal splitting methods with inexact proximal oracles, we are able to appropriately set the tolerance of Jacobi-based methods and gain computation time, while guaranteeing global convergence of the optimization method. Second, we demonstrate using the observations above, and on a variety of problems and splitting algorithms, that the method of Jacobi on a GPU can yield speed-ups of 5-10x in the evaluation of proximal operators of spectral functions, compared to standard approaches on CPU and GPU.

**Related work**   Several approaches have been considered to avoid computing a full eigenvalue or singular value decomposition at each iteration, when the solution is known to be low-rank. Popular approaches include the Lanczos algorithm, see e.g., [CCS10], randomized SVD [HMT11], or nonconvex low-rank formulations [BM03]. However these approaches usually rely on an a priori knowledge of the rank of the solution.

The recent paper [RGN19] focusses on the ADMM method for SDPs and proposes the use of iterative eigensolvers (LOBPCG [Kny01]) to approximately project a symmetric matrix on the positive semidefinite cone. The proposed method can achieve 10x speedups, especially when the primal or dual solution has low-rank. From the theoretical point of view, the approach relies on error bounds derived in [GNR20] concerning approximate projections on the positive semidefinite cone. Our theoretical results in Section 4 constitute a significant generalization of these error bounds to proximal operators of convex spectral functions, and to different notions of approximations.

The papers [CV18] and [JV21] consider proximal splitting algorithms with a generalized Bregman divergence (i.e., different than the squared Euclidean distance), and show that these proximal operators can sometimes be evaluated without explicitly computing a full eigenvalue decomposition. If $\phi$ is a strongly-convex function, the $\phi$-proximal operator of $F$ is defined by

$$\operatorname{prox}_F^\phi(y; a) = \operatorname{argmin}_x \left\{ F(x) + \langle a, x \rangle + \phi(x) - \langle \nabla \phi(y), x \rangle \right\}.$$

The paper [CV18] considers optimization problems over symmetric positive semidefinite Toeplitz matrices and shows that by choosing $\phi$ to be the kernel for the so-called Itakura-Saito distance one can compute the generalized proximal operator in $O(n^2)$ time, instead of $O(n^3)$. In [JV21] it is shown that for sparse semidefinite programs, an adequate choice of $\phi$ leads to a proximal operator that can be computed at the cost of a Cholesky decomposition of a sparse matrix which in general is much smaller than the cost of a full eigenvalue decomposition. Concerning sparse semidefinite programs, there has been a lot of work on devising fast ADMM algorithms using the clique decomposition method [FKMN01], e.g., [ZFP$^+$20, MKL15]. We note that these contributions are specific to sparse semidefinite programs and are complementary to the present paper because they still require an eigenvalue decomposition to be computed at each iteration.

Finally let us mention that there are first-order methods that do not require a full eigenvalue decomposition at each iteration but only the computation of the smallest eigenvalue(s)/eigenvector(s) at each iteration such as e.g., [HR00, Ren14, YTF$^+$21] and for which Lanczos methods are more suited.

**Organization** In Section 2 we recall some necessary background material about convex spectral functions and their proximal operators. In Section 3 we present the Jacobi method to diagonalize symmetric matrices, and explain how it can be used with proximal algorithms. In Section 4 we present the main theoretical result of the paper and show that approximate eigenvalue/singular value decompositions yield guaranteed approximations of proximal operators for spectral functions. Finally we present numerical experiments in Section 5.

## 2   Background

In this section we present some notations and background results concerning spectral functions. If $f : E \to \mathbb{R}$ is a convex function on Euclidean space $E$ we let $\partial f(x)$ be the subdifferential of $f$ at $x$, namely $\partial f(x) = \{g \in E : f(y) \geq f(x) + \langle g, y - x \rangle\}$. An element $g \in \partial f(x)$ is called a subgradient of $f$ at $x$. The *proximal operator* of $f$ at $y$ is defined as

$$\mathrm{prox}_f(y) = \mathrm{argmin}_{x \in E} \left\{ f(x) + (1/2)\|x - y\|^2 \right\}.$$

We let $\mathbf{S}^n$ be the Euclidean space of $n \times n$ symmetric matrices with the trace inner product $\langle X, Y \rangle = \mathrm{tr}(XY)$. We let $\mathbf{S}_+^n$ (resp. $\mathbf{S}_{++}^n$) be the cone of positive semidefinite (resp. definite) matrices in $\mathbf{S}^n$. Also $O(n)$ denotes the set of $n \times n$ orthogonal matrices.

**Spectral functions** A function $F : \mathbf{S}^n \to \mathbb{R}$ of symmetric matrices is called *spectral* if there is a permutation-invariant function $f : \mathbb{R}^n \to \mathbb{R}$ such that $F(X) = f(\lambda(X))$ where $\lambda(X) = (\lambda_1(X), \ldots, \lambda_n(X))$ is the vector of eigenvalues of $X$ (in any order). It is clear that spectral functions are orthogonally-invariant, i.e., $F(U^T X U) = F(X)$, because $U^T X U$ and $X$ have the same eigenvalues; conversely any orthogonally-invariant function $F$ is the spectral function associated to the permutation-invariant function $f(x) = F(\mathrm{diag}(x))$. The main remarkable fact that we will use about spectral functions is a result due to Davis [Dav57], which asserts that $F$ is convex as soon as $f$ is, and gives a variational formula for $F$ certifying this convexity.

**Theorem 1** ([Dav57], see also [BTN01, p.282, claim D]). *Let $f : \mathbb{R}^n \to \mathbb{R}$ be a permutation-invariant function, and let $F : \mathbf{S}^n \to \mathbb{R}$, $F(X) = f(\lambda(X))$ be the associated spectral function. If $f$ is convex on $\mathbb{R}^n$, then $F$ is convex on $\mathbf{S}^n$, and we have the following variational formulation for $F$:*

$$F(X) = \max_{U \in O(n)} f(\mathrm{diag}(U^T X U)). \tag{2}$$

An important, and well-known consequence of the theorem above, is that the proximal operator of $F$ can be obtained from that of $f$, namely for any $Y \in \mathbf{S}^n$ we have

$$\mathrm{prox}_F(Y) = U \, \mathrm{diag}(\mathrm{prox}_f(y)) U^T$$

where $Y = U \, \mathrm{diag}(y) U^T$ is an eigenvalue decomposition of $Y$.

**Functions of singular values** Similar results hold for functions $F : \mathbb{R}^{m \times n} \to \mathbb{R}$ of rectangular matrices that are orthogonally-invariant, i.e., $F(UXV) = F(X)$ for all $U \in O(m)$ and $V \in O(n)$. Any such function can be written as $F(X) = f(\sigma(X))$ where $\sigma(X) \in \mathbb{R}^{\min(m,n)}$ is the vector of singular values and $f : \mathbb{R}^{\min(m,n)} \to \mathbb{R}$ is an *absolutely symmetric* function, i.e., it is invariant under *signed permutations*: $f(\eta_1 x_{\pi(1)}, \ldots, \eta_k x_{\pi(k)}) = f(x_1, \ldots, x_k)$ for $\eta_i = \pm 1$ and $\pi$ a permutation of $\{1, \ldots, k\}$ where $k = \min(m, n)$ [Lew95]. In this case too, it holds that $F$ is convex as soon as $f$ is, and we have:

$$F(X) = \max_{U \in O(m), V \in O(n)} f(\mathrm{diag}(U^T X V)). \tag{3}$$

The variational formulas (2) and (3) are a consequence of well-known results concerning majorization relations between the diagonal of a matrix and its eigenvalues or singular values [MOA79, Chapter 9]. For completeness we have included a proof of these formulas in the supplementary material. We refer the reader to [PB14, Section 6.7] and references therein for more details on the proximal operator of spectral and orthogonally-invariant functions.

# 3 Jacobi-based methods for eigenvalue and singular value decompositions

In this section we present Jacobi-based methods to compute eigenvalue and singular value decompositions, and explain how they can be used in proximal optimization algorithms. For concreteness we will focus on the eigenvalue decomposition of symmetric matrices, but as will be clear, the extension to the SVD of rectangular matrices will be straightforward.

## 3.1 The method

Jacobi eigenvalue methods compute the eigenvalue decomposition of a real symmetric matrix by iteratively eliminating off-diagonal elements using *Givens rotations*. More precisely, let $A$ be a $n \times n$ real symmetric matrix that we want to diagonalize. The Jacobi method begins by selecting an off-diagonal index $(i, j)$, with $i < j$, such that $|A_{ij}| > 0$, and constructs a $2 \times 2$ rotation matrix $\left[ \begin{smallmatrix} \cos\theta & \sin\theta \\ -\sin\theta & \cos\theta \end{smallmatrix} \right]$ that diagonalizes the $2 \times 2$ submatrix $\left[ \begin{smallmatrix} A_{ii} & A_{ij} \\ A_{ij} & A_{jj} \end{smallmatrix} \right]$. The rotation matrix can be embedded in a larger $n \times n$ Givens rotation matrix at indices $(i, j)$ which is otherwise equal to the identity matrix:

$$\begin{cases} G_{ii} = G_{jj} = \cos\theta \\ G_{ij} = \sin\theta, G_{ji} = -\sin\theta \end{cases} \quad \text{and} \quad G_{pq} = \delta_{p,q} \quad \text{for } p \notin \{i, j\} \text{ or } q \notin \{i, j\},$$

where $\delta_{p,q} = 1$ if $p = q$ and 0 otherwise. This Givens rotation matrix satisfies $(GAG^T)_{ij} = 0$. The Jacobi algorithm then proceeds by updating $A \leftarrow GAG^T$ and performs the same process until a convergence criterion is satisfied. Usually the convergence criterion is given in terms of $\mathrm{off}(A)$, defined to be the square root of the sum of the off-diagonal elements of $A$:

$$\mathrm{off}(A) := \sqrt{\sum_{1 \leq i \neq j \leq n} A_{ij}^2}. \tag{4}$$

The algorithm can thus be summarized as follows:

---

**Algorithm 1:** Jacobi method for diagonalizing symmetric matrix $A$

---

**while** *off*$(A) \geq$ *tolerance* **do**
    choose an off-diagonal index $(i, j)$ such that $|A_{ij}| > 0$
    construct a Givens matrix $G$ such that $(GAG^T)_{ij} = 0$
    update $A \leftarrow GAG^T$
**end**

---

In the classical Jacobi method, the off-diagonal entry $(i, j)$ selected at each iteration is the one for which $|A_{ij}|$ is largest. It can be easily shown that with this choice, the quantity $\mathrm{off}(A)$ satisfies after one iteration

$$\mathrm{off}(GAG^T)^2 \leq (1 - 1/N)\,\mathrm{off}(A)^2$$

where $N = n(n-1)/2$. As such $\mathrm{off}(A)$ converges linearly to 0. It has been shown by Schönhage, and van Kempen [Sch64, VK66a] that the method actually attains quadratic convergence locally.

**Cyclic method** The classical Jacobi method requires searching for a largest element among the off-diagonal elements at each iteration. For an $n \times n$ matrix the cost of this step is $\mathcal{O}(n^2)$ flops. Cyclic Jacobi methods instead cycle through all off-diagonal elements of the matrix in some fixed order, computing the associated Givens matrices and updating accordingly. The algorithm proceeds by *sweeps*, where in each sweep each off-diagonal entry is processed exactly once. Various convergence results have been proved for cyclic methods most notably local quadratic convergence, see e.g., [FH60, Wil62, VK66b]. In practice, the number of sweeps needed to diagonalize a matrix of size $n \lesssim 1000$ is usually no more than 15.

**Parallelization** The cyclic Jacobi method can be easily parallelized. Indeed, instead of eliminating a single off-diagonal entry at each iteration, one can eliminate in parallel $\lfloor n/2 \rfloor$ off-diagonal entries that correspond to distinct rows/columns. More precisely by selecting indices $(i_1, j_1), \ldots, (i_m, j_m)$

where $m = \lfloor n/2 \rfloor$ such that $i_1, j_1, i_2, j_2, \ldots, i_m, j_m$ are all distinct one can compute in parallel the $m$ Givens rotations that are needed to eliminate the entries $A_{i_1 j_1}, \ldots, A_{i_m j_m}$. A single sweep can then be performed in $n-1$ steps where at each step $k = 1, \ldots, n-1$ one chooses indices $(i_1[k], j_1[k]), \ldots, (i_m[k], j_m[k])$ that are pairwise distinct in such a way that all the off-diagonal entries are processed at least once. For more details on the choice of these indices we refer the reader to [GVL13, Section 8.5].

**Other eigenvalue algorithms** In general, the Jacobi method is not competitive with more traditional eigenvalue algorithms such as the QR method or the divide-and-conquer method. Counting floating point operations only, the QR method is roughly equivalent to two sweeps of Jacobi [GVL13, TBI97]. One important difference between the Jacobi method and QR/divide-and-conquer algorithms is that the latter start by putting the matrix $A$ into tridiagonal form by a sequence of orthogonal (Householder) transformations, whereas the Jacobi method operates on $A$ directly. One consequence of this is that the Jacobi method can have faster convergence if $\mathrm{off}(A)$ is small, whereas this is not the case of other algorithms that start by a reduction into tridiagonal form. This ability to exploit an approximate diagonalization of $A$ is what makes the Jacobi method particularly interesting for iterative optimization algorithms.

**Singular value decomposition** The Jacobi algorithm can be very easily adapted to compute the singular value decomposition of rectangular matrices. The idea is to solve a sequence of $2 \times 2$ SVD problems, instead of $2 \times 2$ symmetric eigenvalue problems. This leads to the two-sided Jacobi SVD algorithm. Another approach, called the one-sided Jacobi SVD algorithm, performs orthogonal transformations on the right to make the columns of $A$ orthogonal, by iteratively orthogonalizing pairs of columns. From there, one can extract the singular values as the Euclidean norms of the columns of $A$, and the left singular vector matrix easily. We refer the reader to [GVL13, Section 8.6.4] for more details and references.

### 3.2   Jacobi-based methods in proximal algorithms

We now explain how the Jacobi method can be used in iterative optimization algorithms for the computation of proximal operators of spectral functions. The main idea is to use the eigenvector matrix from the previous iteration as a warm-start for the current iterate. More precisely, assume that at iteration $k$ one needs to compute the proximal operator $\mathrm{prox}_F(Y_k)$ where $F : \mathbf{S}^n \to \mathbb{R}$ is the spectral function associated to the permutation-invariant function $f : \mathbb{R}^n \to \mathbb{R}$. We proceed as follows:

- Let $U_{k-1}$ be the orthogonal eigenvector matrix computed at iterate $k-1$
- Apply the Jacobi method to $U_{k-1}^T Y_k U_{k-1}$ with tolerance $\epsilon_k$. This returns an orthogonal matrix $W$ such that
$$\mathrm{off}(W^T U_{k-1}^T Y_k U_{k-1} W) \le \epsilon_k$$
- Set $U_k = U_{k-1} W$ (eigenvector matrix of $Y_k$) and $y_k = \mathrm{diag}(U_k^T Y_k U_k)$ (eigenvalues of $Y_k$)
- Output $U_k \, \mathrm{diag}(\mathrm{prox}_f(y_k)) U_k^T$ as the proximal operator $\mathrm{prox}_F(Y_k)$.

As we demonstrate in the numerical experiments section below, a very small number of sweeps (often just one) are in general needed to converge to the eigenvalue decomposition of $Y_k$ in the second step above, because the matrix $U_{k-1}^T Y_k U_{k-1}$ is usually very close to being diagonal, especially after the first few iterations of the optimization algorithm.

It is straightforward to extend the warm-start approach described here to the situation where we want to compute singular value decompositions, and we omit the details for brevity.

**Remark 1.** *When numerically implementing the procedure above, the accumulation of errors can cause the columns of $U_k$ not to be normalized after a large number of iterations. To resolve this issue one can orthonormalize the columns of $U_k$ periodically using a QR decomposition.*

## 4   Approximate proximal evaluation

Most proximal algorithms (e.g., ADMM, proximal gradient method, ...) are still convergent even if only an approximation of the proximal operator is computed at each iteration, provided the approxi-

mation quality suitably decays to zero, see e.g., [EB92, SRB11, Yao16, BTB20]. A natural question is to understand how the tolerance parameter of the Jacobi method $\epsilon_k$ affects the approximation quality of the proximal operator $\text{prox}_F(Y_k)$? This section addresses this question.

**Measures of quality of prox** There are multiple ways to quantify the approximation quality of a proximal operator, see e.g., [BTB20] which compiles a list of different ways that were considered in the literature. In this section we will focus on two simple notions that have been used in convergence guarantees for proximal splitting methods. Let $F : E \to \mathbb{R}$ be a convex function for which we want to compute the prox at a point $Y \in E$. Let

$$\Phi(X; Y) = F(X) + (1/2)\|X - Y\|_F^2$$

be the prox objective so that $\text{prox}_F(Y) = \text{argmin}_{X \in E} \Phi(X; Y)$. If $\hat{X} \in E$ is a candidate value for $\text{prox}_F(Y)$ the two measures of approximations we consider are: (i) Euclidean distance to the true prox, i.e., $\delta = \|\hat{X} - \text{prox}_F(Y)\|_F$ and (ii) suboptimality in prox objective value, i.e., $\eta = \Phi(\hat{X}; Y) - \Phi(\text{prox}_F(Y); Y)$. The former notion was used e.g., to guarantee convergence of ADMM algorithms, assuming that the sequence of errors $(\delta_k)$ is summable [EB92, Yao16]; and the latter was used in [SRB11] to guarantee convergence of (fast) proximal gradient methods, under a decaying assumption on $(\eta_k)$.

**Remark 2.** *Note that since $\Phi$ is 1-strongly convex we always have the inequality:*

$$\Phi(\hat{X}; Y) - \Phi(\text{prox}_F(Y); Y) \geq \frac{1}{2}\|\hat{X} - \text{prox}_F(Y)\|^2. \tag{5}$$

*This means that any bound on the approximation quality in terms of prox objective value, implies a bound on the approximation quality using the Euclidean norm.*

**Approximate prox using approximate eigenvalue/singular decomposition** Our main result gives bounds on the quality of the prox of any convex spectral function computed using an approximate eigenvalue decomposition. More precisely, assume that $F : \mathbf{S}^n \to \mathbb{R}$ is a convex spectral function of symmetric matrices associated to the permutation-invariant function $f(x_1, \ldots, x_n)$. Let $Y \in \mathbf{S}^n$ for which we want to compute $\text{prox}_F(Y)$. We know that $\text{prox}_F(Y) = U \text{prox}_f(y) U^T$ where $y = (y_1, \ldots, y_n)$ is the vector of eigenvalues of $Y$, and $U$ is the orthogonal matrix that diagonalizes $Y$. Assume that $\hat{U}$ is an orthogonal matrix such that $\text{off}(\hat{U}^T Y \hat{U}) \leq \epsilon$, i.e.,

$$\hat{U}^T Y \hat{U} = \text{diag}(\hat{y}) + E \quad \text{where} \quad \begin{cases} \|E\|_F \leq \epsilon \\ E_{ii} = 0 \text{ for } i = 1, \ldots, n. \end{cases} \tag{6}$$

Our main theoretical result, stated below, quantifies the error in the approximation $\hat{X} \approx \text{prox}_F(Y)$ where

$$\hat{X} = \hat{U} \text{diag}(\text{prox}_f(\hat{y})) \hat{U}^T. \tag{7}$$

**Theorem 2.** *Under the assumptions above, we have:*
*(i)* $\|\hat{X} - \text{prox}_F(Y)\|_F \leq \epsilon$; *and*
*(ii)* $\Phi(\hat{X}; Y) - \Phi(\text{prox}_F(Y); Y) \leq \epsilon^2/2$.

A similar result can be stated for orthogonally-invariant functions of rectangular matrices.

**Theorem 3.** *Let $F : \mathbb{R}^{m \times n} \to \mathbb{R}$ be a convex orthogonally-invariant function associated with the absolutely-symmetric function $f(x) = F(\text{diag}(x))$ defined on $\mathbb{R}^{\min(m,n)}$. Let $Y \in \mathbb{R}^{m \times n}$, and assume that $\hat{U} \in O(m)$ and $V \in O(n)$ are orthogonal matrices such that*

$$\hat{U}^T Y \hat{V} = \text{diag}(\hat{y}) + E \quad \text{where} \quad \begin{cases} \|E\|_F \leq \epsilon \\ E_{ii} = 0 \text{ for } i = 1, \ldots, \min(m, n). \end{cases}$$

*Then, with $\hat{X} = \hat{U} \text{diag}(\text{prox}_f(\hat{y})) \hat{V}^T$, the following holds:*
*(i)* $\|\hat{X} - \text{prox}_F(Y)\|_F \leq \epsilon$, *and*
*(ii)* $\Phi(\hat{X}; Y) - \Phi(\text{prox}_F(Y); Y) \leq \epsilon^2/2$.

The proofs of both theorems are in the supplementary material. Some remarks are in order:

- Inequality (i) generalizes the main result in [GNR20].[1] The main result in [GNR20] shows that when $F$ is the indicator function of the positive semidefinite cone (so that $\text{prox}_F$ is the Euclidean projection on $\mathbf{S}_+^n$) then $\|\hat{X} - \text{prox}_F(Y)\|_F \leq \sqrt{2}\epsilon$. Our result significantly generalizes the latter to the proximal operator of arbitrary convex spectral functions $F$.

- Note that inequality (i) actually follows from (ii) by (5).

We believe Theorems 2 and 3 are remarkable in a number of ways:

- They bound the distance to the true prox (a quantity that is hard to compute) using a simple quantity which is $\epsilon = \text{off}(\hat{U}^T Y \hat{U})$.

- the bounds apply to *all* convex spectral functions $F$ and do not depend on any Lipschitz assumption on $F$;

- the bounds do not depend on how the orthogonal matrix $\hat{U}$ is obtained; in this paper we consider the Jacobi method for obtaining approximate diagonalizations, however the bounds can be used with any other method;

- well-known perturbation bounds for eigenvalue/eigenvector computation (e.g., the Davis-Kahan theorem) suggest that the error would depend on additional factors related to the spectrum of $Y$. Such a dependence would make the theorem very hard to apply. Our bounds are free of such dependence and only depend on $\epsilon$.

**Implications** If we combine our theorems with [EB92] and [SRB11] we get guarantees on the convergence rates of ADMM and (fast) proximal gradient methods, when the proximal operators are evaluated using an approximate eigenvector matrix. To illustrate this, consider for instance the following optimization over symmetric matrices

$$\min_{X \in \mathbf{S}^n} F(X) + G(X) \tag{8}$$

where $F(X)$ is a convex spectral function and $G(X)$ is convex and smooth with $L$-Lipschitz gradient. Consider the (accelerated) proximal gradient method

$$X_{k+1} = \text{prox}_{L^{-1}F}(Z - (1/L)\nabla G(Z)) \tag{9}$$

where $Z = X_k$ for the standard proximal gradient method, and $Z = X_k + \frac{k-1}{k+2}(X_k - X_{k-1})$ for the accelerated version. Assume that the proximal operator in (9) is evaluated approximately according to (6)-(7) with accuracy $\epsilon_k$. Then, using [SRB11, Props. 1 and 2] and Theorem 2(ii), we immediately get: (a) if $(\epsilon_k)$ is summable (e.g., $\epsilon_k = O(1/k^{1+\delta})$) then the (basic) proximal gradient method converges at a rate $O(1/k)$ to the optimal value of (8); and (b) if $(k\epsilon_k)$ is summable (e.g., $\epsilon_k = O(1/k^{2+\delta})$) then the accelerated proximal gradient method converges at a rate $O(1/k^2)$ to the optimal value of (8). For the ADMM algorithm, we can combine the result of [EB92, Theorem 8] with Theorem 2(i) to guarantee convergence when $(\epsilon_k)$ is summable.

## 5   Numerical experiments

In this section we present numerical experiments with Jacobi-based methods on different problems and splitting algorithms.

**Setup** We use the parallel version of the Jacobi method implemented on GPU, and available through the NVIDIA CUSOLVER library [NVI20] in the functions `syevj` and `gesvdj` for symmetric eigenvalue decomposition and singular value decomposition respectively. We compare it with the standard functions on both CPU/GPU implemented in LAPACK/CUSOLVER's functions `syevd` and `gesvd`. All our code is in Python and the latter functions are invoked respectively by NumPy's [HMvdW+20] and CuPy's [OUN+17] functions `linalg.eigh` and `linalg.svd`. The GPU we used is a NVIDIA TITAN Xp with CUDA 11.0, and the CPU is an Intel Core i5-6500 with 4 cores at 3.20 GHz and 32GB RAM, running Ubuntu 18.04. For GPU computations (default eigenvalue solver and Jacobi's method) we use 32-bit floats, instead of 64-bit doubles, which are much faster on GPU. On the examples we have tried this had little or no effect on the quality of the solutions, especially because we are looking at problems where moderate accuracy is satisfactory.

---

[1]We note that the proof of inequality (i) does not actually use the assumption that $E_{ii} = 0$ for $i = 1, \ldots, n$.

**Performance of Jacobi's method on GPU**   Before showing the results on optimization problems, we first compare in Table 1 the time needed for a single Jacobi sweep vs. the time needed for a full eigenvalue (resp. singular value) decomposition using NumPy/CuPy's `linalg.eigh` (resp. `linalg.svd`), on a random Gaussian $n \times n$ matrix. For symmetric eigenvalue decompositions, we note that for matrices of size $\sim 500$, one sweep of Jacobi is about 10x faster than the default methods. However the default method becomes faster for matrices of size $\gg 2000$.[2] On the other hand, for singular value decomposition the ratio is more pronounced and attains 40x for matrices of size $\gtrsim 2000$.

| | Symmetric eigenvalue decomposition | | | Singular value decomposition | | |
|---|---|---|---|---|---|---|
| $n$ | default (CPU) | default (GPU) | **jacobi-1sw (GPU)** | default (CPU) | default (GPU) | **jacobi-1sw (GPU)** |
| 128 | 2.18 | 5.91 | **0.83** | 3.75 | 12.23 | **1.14** |
| 256 | 6.86 | 12.72 | **1.43** | 12.93 | 39.93 | **1.85** |
| 512 | 26.24 | 31.78 | **3.29** | 62.46 | 107.47 | **3.99** |
| 1024 | 152.44 | 63.28 | **11.45** | 385.59 | 476.22 | **15.17** |
| 2048 | 1180.78 | 163.62 | **59.94** | 3,453.76 | 3,223.60 | **77.18** |
| 4096 | 8491.13 | 532.92 | **682.72** | 24,869.11 | 19,649.37 | **500.99** |

Table 1: Times, in milliseconds, for one sweep of the Jacobi method on a GPU vs. full eigenvalue/singular-value decomposition using the default method on CPU/GPU.

## 5.1   Sparse inverse covariance selection

We consider here the well-known statistical problem of estimating an unknown covariance matrix $\Sigma \in \mathbf{S}_{++}^n$ from samples $x_1, \ldots, x_N \in \mathbb{R}^n$ of a Gaussian distribution $\mathcal{N}(0, \Sigma)$, under the assumption that $\Sigma^{-1}$ is sparse. The approach proposed in [BGdN06, FHT08] consists of solving the following convex optimization problem:

$$\min_{X \in \mathbf{S}_{++}^n} \quad \mathrm{tr}(\hat{S}X) - \log \det X + \lambda \|X\|_1 \tag{10}$$

where $\lambda > 0$ is a regularization parameter, $\hat{S} \in \mathbf{S}_+^n$ is the given empirical covariance matrix, and $\|X\|_1 = \sum_{ij} |X_{ij}|$ is the entrywise $\ell_1$-norm of $X$. The ADMM algorithm is one of the most popular methods for this problem, see e.g., [SMG10, BPC11] and references therein. The problem (10) can be naturally put in the splitting form (1) with $F(X) = \mathrm{tr}(\hat{S}X) - \log \det X$ and $G(X) = \lambda \|X\|_1$. The ADMM iterates can be written as:

$$\begin{cases} X_{k+1} = \mathrm{prox}_{-\rho \log \det}(Z_k - U_k - \rho \hat{S}) \\ Z_{k+1} = \mathsf{S}_{\lambda/\rho}(X_{k+1} + U_k) \\ U_{k+1} = U_k + X_{k+1} - Z_{k+1} \end{cases} \tag{11}$$

where the proximal operator of $-\rho \log \det$ is $\mathrm{prox}_{-\rho \log \det}(Y) = V \operatorname{diag}\left((y + \sqrt{y^2 + 4\rho})/2\right) V^T$ where $Y = V \operatorname{diag}(y) V^T$ is the eigenvalue decomposition of $Y$; and $\mathsf{S}_t$ is the entrywise soft-thresholding operation defined by $\mathsf{S}_t(x) = 0$ if $|x| \leq t$ and $x - \mathrm{sign}(x)t$ otherwise. The main computational cost in the iterations (11) is the proximal update rule for $X_k$ which requires the eigenvalue decomposition of $Z_k - U_k - \rho \hat{S}$ at each iteration. We can use the Jacobi method to compute this eigenvalue decomposition, using the previous iterate's orthogonal transformation as a warm start, as suggested in Section 3.2.

We have implemented this approach with different values of $n$ in the range 200-1000, where the covariance matrix $\Sigma \in \mathbf{S}_{++}^n$ was chosen to be a banded matrix with constant bandwidth $k = 5$. We have used the parameters and stopping criterion of the Matlab implementation of (11) from [BPC11] available at `https://web.stanford.edu/~boyd/papers/admm/`. We compare the Jacobi method

---

[2]It is not clear to us why the performance of the Jacobi-symmetric eigenvalue routine `syevj` significantly degrades for sizes $\gg 2000$. Note that, surprisingly, the Jacobi-SVD routine on matrices of size 4096 is *faster* than the Jacobi-symmetric eigenvalue routine on the same size.

with the standard routines in linear algebra packages for eigenvalue decomposition on CPU and GPU (NumPy/CuPy's `linalg.eigh`).

The results of the numerical experiments are shown in Table 2. Depending on the size of the matrix, the Jacobi method can be more than 5x faster than the standard approaches on both CPU and GPU. For the Jacobi method we used a relative tolerance that decays like $1/k^2$, more precisely we set $\epsilon_k = \|Y_k\|_F/(1 + k^2)$, where $Y_k$ is the matrix to diagonalize at iteration $k$. We see that the average number of sweeps per iteration is $\approx 1$ confirming the fact that the eigenvector matrix from iterate $k - 1$ is a good initial point for the $k$'th iterate.

| | iterations | | | $X$-update time per iter (ms) | | | |
|---|---|---|---|---|---|---|---|
| $n$ | default (CPU) | default (GPU) | jacobi (GPU) | default (CPU) | default (GPU) | **jacobi (GPU)** | sweeps/iter |
| 200 | 308 | 308 | 309 | 6.20 | 7.43 | **1.81** | 0.97 |
| 400 | 269 | 269 | 270 | 25.83 | 19.56 | **3.06** | 0.97 |
| 600 | 248 | 248 | 249 | 62.64 | 33.41 | **5.62** | 0.98 |
| 800 | 236 | 236 | 237 | 100.06 | 46.55 | **10.7** | 0.99 |
| 1000 | 227 | 227 | 227 | 234.32 | 67.42 | **16.4** | 0.99 |

Table 2: Sparse covariance selection problem. Time per iteration for the computation of the $X$-proximal update in (11) using the Jacobi method compared to the standard approaches on CPU/GPU.

## 5.2 Semidefinite programming

We consider here semidefinite programs in primal-dual form,

$$
\begin{aligned}
\min_{x \in \mathbb{R}^n, S \in \mathbf{S}^k} \quad & c^T x & \max_{Y \in \mathbf{S}^k} \quad & -\operatorname{tr}(BY) \\
\text{s.t.} \quad & \sum_{i=1}^n x_i A_i + S = B,\ S \succeq 0 & \text{s.t.} \quad & Y \succeq 0,\ \langle A_i, Y \rangle = -c_i\ (i = 1, \dots, n)
\end{aligned}
\tag{12}
$$

where $c \in \mathbb{R}^n$, $A_1, \dots, A_n \in \mathbf{S}^k$ and $B \in \mathbf{S}^k$. A primal-dual ADMM-based algorithm was proposed in [OCPB16] and implemented in the SCS package [OCPB19]. Each iteration of the algorithm consists of two steps, one projection onto a linear subspace, and one projection onto the positive semidefinite cone $\mathbf{S}_+^k$. The projection step can be computed using the Jacobi-based method. Table 3 shows the time per iteration required for the projection step using Jacobi's method compared to the default approaches, on random instances of (12). Like in the previous example, we used a relative tolerance for the Jacobi method that decays like $1/k^2$, more precisely, $\epsilon_k = \|Y_k\|_F/(1 + k^2)$, where $Y_k$ is the matrix to diagonalize at iteration $k$. We see that the time spent in the computation of the projection on the positive semidefinite is up to 5x faster than the default approaches, and that the number of sweeps per iteration is close to 1.

| | iterations | | | proj time/iter (ms) | | | | linsolve/iter (ms) |
|---|---|---|---|---|---|---|---|---|
| $k$ | default (CPU) | default (GPU) | jacobi (GPU) | default (CPU) | default (GPU) | **jacobi (GPU)** | sweeps per iter | |
| 200 | 60 | 60 | 60 | 5.28 | 8.01 | **11.63** | 1.00 | 2.46 |
| 400 | 80 | 80 | 80 | 21.75 | 20.57 | **3.70** | 1.01 | 10.07 |
| 600 | 90 | 90 | 90 | 66.05 | 32.62 | **7.97** | 1.01 | 25.76 |
| 800 | 90 | 90 | 90 | 95.20 | 47.9 | **11.65** | 1.01 | 39.10 |
| 1000 | 150 | 150 | 150 | 247.76 | 74.24 | **18.55** | 1.01 | 58.76 |

Table 3: Semidefinite programming using primal-dual ADMM approach of [OCPB16] on random instances of (12) where $n = 3k$, $B = I_k$, and the $A_i$ are sparse random matrices with density $\sim k$. Time per iteration (in milliseconds) to compute projection of $k \times k$ symmetric matrix on the positive semidefinite cone using the Jacobi method vs. standard approaches on CPU/GPU. For comparison, we also show the time needed for the linear system solve at each iteration (in milliseconds). We used the same termination criterion as the one in [OCPB16], with a tolerance of $10^{-3}$ on the normalized primal and dual residuals, and duality gap.

## 5.3 Nuclear norm minimization using the proximal gradient method

We consider the low-rank matrix recovery problem

$$\min_{X \in \mathbb{R}^{n \times n}} (1/2)\|\mathcal{A}(X) - b\|_2^2 + \lambda\|X\|_* \tag{13}$$

where $\mathcal{A} : \mathbb{R}^{n \times n} \to \mathbb{R}^m, b \in \mathbb{R}^m$, $\lambda$ is the regularization parameter, and $\|X\|_* = \sum_{i=1}^n \sigma_i(X)$ is the nuclear norm. A popular method for solving (13) is the (fast) proximal gradient method, with iterates:

$$X_{k+1} = \mathsf{SVT}_{t\lambda}(Z - t\mathcal{A}^*(\mathcal{A}(Z) - b)) \tag{14}$$

where $t > 0$ is the step size, and $\mathsf{SVT}_{t\lambda}$ is the proximal operator associated to the nuclear norm which is given by the soft-thresholding of singular values. In the standard proximal gradient method, we take $Z = X_k$ in (14), whereas for the fast proximal gradient method we take $Z = X_k + \frac{k-1}{k+2}(X_k - X_{k-1})$. Clearly, the main computational cost in the implementation of (14) is to compute the singular value decomposition of a $n \times n$ matrix.

Table 4 shows the result of using the fast proximal gradient method with the Jacobi-based method vs. the default SVD routines on CPU/GPU for the computation of the proximal operator, on random matrix-completion instances of (13). Like for the previous examples, we used a relative tolerance for the Jacobi method decaying like $1/k^2$, except for the largest instance where we fixed the number of sweeps to be 1 at each iteration. The number of iterations (14) was fixed and set to 1000, with a time limit of 1 hour on the total running time. For the largest instance the two default methods exceeded the time limit and the objective value we report is the one obtained after 1000 iterations.

| | objective | | | SVD time per iter (ms) | | | |
|---|---|---|---|---|---|---|---|
| $n$ | default CPU | default GPU | jacobi GPU | default CPU | default GPU | **jacobi GPU** | jacobi sweeps per iter |
| 200 | 7.64e+01 | 7.64e+01 | 7.64e+01 | 7.1 | 16.8 | **2.1** | 1.4 |
| 400 | 3.15e+02 | 3.15e+02 | 3.15e+02 | 29.4 | 53.1 | **4.9** | 1.9 |
| 600 | 7.24e+02 | 7.24e+02 | 7.24e+02 | 71.1 | 105.4 | **10.1** | 2.3 |
| 800 | 1.29e+03 | 1.29e+03 | 1.29e+03 | 149.5 | 177.7 | **17.4** | 2.5 |
| 1000 | 1.99e+03 | 1.99e+03 | 1.99e+03 | 306.2 | 327.7 | **29.3** | 2.6 |
| 4000 | 4.65e+04* | 3.94e+04* | 3.41e+04 | 22,641.5 | 10,676.7 | **597.5** | 1.0** |

Table 4: Nuclear norm minimization using fast proximal gradient method (14). Time to compute the SVD at each iteration using the Jacobi's method vs. the default methods on CPU/GPU. Objective value is after 1000 iterations (except for $n = 4000$, default CPU/GPU).
*Objective value after the time limit of 1h was reached.
**The number of sweeps of the Jacobi method was fixed to 1 for the instance $n = 4000$.

## Acknowledgments

H.G. is supported by a GlaxoSmithKline PhD studentship. H.F. would like to thank Oisín Faust and Alhussein Fawzi for comments on an earlier draft of this paper.

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
