# Supplementary material for the paper "*Faster proximal algorithms for matrix optimization using Jacobi-based eigenvalue methods*"

Hamza Fawzi        Harry Goulbourne

Department of Applied Mathematics and Theoretical Physics
University of Cambridge, UK
{hf323, hmg42}@cam.ac.uk

In this document we give the proof for Theorems 2 and 3 in the paper. The proofs of these theorems rely on variational formulas (2) and (3) for convex spectral and orthogonally-invariant functions. We include a proof of the latter for completeness, and because we could not find an explicit reference with (3) in the literature.

## 1    Proof of variational formulas for convex spectral and orthogonally-invariant functions

### 1.1    Symmetric case

We prove the following theorem.

**Theorem 1.** *Let $f : \mathbb{R}^n \to \mathbb{R}$ be a convex permutation-invariant[1] function and let $F : \mathbf{S}^n \to \mathbb{R}$ be the associated spectral function $F(X) = f(\lambda(X))$. Then for any $X \in \mathbf{S}^n$ we have:*

$$F(X) = \max_{U \in O(n)} f(\mathrm{diag}(U^T X U)). \tag{1}$$

Note that (1) proves that $F$ is convex, as a pointwise maximum of convex functions, since $X \mapsto f(\mathrm{diag}(U^T X U))$ is convex for any fixed $U$. A proof of (1) appears in [BTN01, p.282, claim D]. We include a short proof here for completeness, and because similar ideas will be used for the rectangular case (below) for which we could not find an explicit reference in the literature.

The proof of (1) relies on the following two results. Recall that a matrix $P \in \mathbb{R}^{n \times n}$ is *doubly stochastic* if $P_{ij} \geq 0$ for all $1 \leq i, j \leq n$ and $\sum_{i=1}^n P_{ij} = \sum_{i=1}^n P_{ji} = 1$ for all $j = 1, \ldots, n$.

**Lemma 1.** *If $Y \in \mathbf{S}^n$, then there is a doubly stochastic matrix $P \in \mathbb{R}^{n \times n}$ such that $\mathrm{diag}(Y) = P\lambda(Y)$, where $\lambda(Y) \in \mathbb{R}^n$ is the vector of eigenvalues of $Y$ (in any order).*

*Proof.* Write $Y = V \mathrm{diag}(\lambda(Y))V^T$ with $V$ orthogonal. Then $Y_{ii} = \sum_j (V_{ij})^2 \lambda(Y)_j$, i.e., $\mathrm{diag}(Y) = P\lambda(Y)$ with $P = (V_{ij}^2)$. Since $V$ is orthogonal, $P$ is doubly-stochastic. $\square$

**Remark 1.** *Lemma 1 says that $\mathrm{diag}(Y)$ is majorized by $\lambda(Y)$.*

**Theorem 2** (Birkhoff-von Neumann)**.** *Any doubly-stochastic matrix is the convex combination of permutation matrices.*

The proof of (1) is now straightforward. The inequality $\leq$ is clear (simply take $U$ to be the eigenvector matrix for $X$.) For the inequality $\geq$, let $U \in O(n)$. By Lemma 1, there is a doubly-stochastic matrix

---

[1]i.e., $f(x_{\sigma(1)}, \ldots, x_{\sigma(n)}) = f(x_1, \ldots, x_n)$ for all $x \in \mathbb{R}^n$ and permutations $\sigma$ of $\{1, \ldots, n\}$

35th Conference on Neural Information Processing Systems (NeurIPS 2021).

$P$ such that $\mathrm{diag}(U^T X U) = P\lambda(U^T X U) = P\lambda(X)$. Write $P = \sum_i \alpha_i S_i$ for permutation matrices $S_i$, where $\sum_i \alpha_i = 1$ and $\alpha_i \geq 0$. Then

$$f(\mathrm{diag}(U^T X U)) = f(P\lambda(X)) = f(\sum_i \alpha_i S_i \lambda(X))$$

$$\overset{(*)}{\leq} \sum_i \alpha_i f(S_i \lambda(X)) \overset{(**)}{=} \sum_i \alpha_i f(\lambda(X)) = f(\lambda(X)) = F(X),$$

where we used the fact that $f$ is convex in $(*)$ and that it is permutation-invariant in $(**)$. This proves (1).

## 1.2 Rectangular case

We now prove a similar variational formula for functions of singular values.

**Theorem 3.** *Let $f$ be a convex* absolutely symmetric function[2] *on $\mathbb{R}^m$. Let $F : \mathbb{R}^{m \times n} \to \mathbb{R}$ with $m \leq n$ be the associated orthogonally-invariant function, i.e., $F(X) = f(\sigma(X))$ where $\sigma(X) \in \mathbb{R}^m$ is the vector of singular values of $X$. Then the following holds, for any $X \in \mathbb{R}^{m \times n}$:*

$$F(X) = \max_{U \in O(m), V \in O(n)} f(\mathrm{diag}(U^T X V)). \tag{2}$$

Formula (2) implies that $F$ is convex, as the pointwise maximum of convex functions. We could not find a complete proof of (2) in the literature, even though it follows from existing well-known facts about majorization relations between the diagonal elements of a matrix, and its vector of singular values [MOA79, Chapter 9]. We thus include a proof for completeness.

The proof of (2) follows the same lines as the proof of the symmetric case. We need the following definitions.

**Definition 1.** *A matrix $P \in \mathbb{R}_+^{m \times m}$ is called* substochastic *if $P_{ij} \geq 0$ for all $1 \leq i, j \leq m$ and $\sum_{i=1}^m P_{ij} \leq 1$ and $\sum_{i=1}^m P_{ji} \leq 1$ for all $j = 1, \ldots, m$. A matrix $S \in \mathbb{R}^{m \times m}$ is called a* subpermutation *matrix if the entries of $S$ are either 0 or 1, and each row and column has* at most *one nonzero entry.*

**Lemma 2.** *If $Y \in \mathbb{R}^{m \times n}$ with $m \leq n$, there there is a* substochastic *matrix $P \in \mathbb{R}_+^{m \times m}$ such that $|\mathrm{diag}(Y)| = P\sigma(Y)$.*

**Remark 2.** *This lemma says that the vector $|\mathrm{diag}(Y)|$ is* weakly majorized *by the vector $\sigma(Y)$.*

*Proof.* The proof we give here is from [MOA79, Theorem 9.D.1]. Consider the singular value decomposition of $Y$: $Y = U\Sigma V^T$ where $U \in O(m)$ and $V \in O(n)$ with $\Sigma = \mathrm{diag}(\sigma)$. Then $Y_{ii} = \sum_{j=1}^m U_{ij} V_{ij} \sigma_j$ and so $|Y_{ii}| \leq \sum_{j=1}^m |U_{ij} V_{ij}| \sigma_j = (\tilde{P}\sigma)_i$ where $\tilde{P}_{ij} = |U_{ij} V_{ij}|$. Note that $\tilde{P}$ is substochastic because

$$\sum_{i=1}^m \tilde{P}_{ij} = \sum_{i=1}^m |U_{ij} V_{ij}| \leq \frac{1}{2} \sum_{i=1}^m U_{ij}^2 + \frac{1}{2} \sum_{i=1}^m V_{ij}^2 \leq 1$$

since $U$ and $V$ are orthogonal matrices (one proves that $\sum_{j=1}^m \tilde{P}_{ij} \leq 1$ in the same way). Since $|Y_{ii}| \leq (\tilde{P}\sigma)_i$ we can write $|Y_{ii}| = d_i(\tilde{P}\sigma)_i$ where $0 \leq d_i \leq 1$, and so $|\mathrm{diag}(Y)| = P\sigma$ where $P = \mathrm{diag}(d)\tilde{P}$. Since $\tilde{P}$ is substochastic, $P = \mathrm{diag}(d)\tilde{P}$ is also substochastic. This completes the proof. $\square$

**Lemma 3.** *Any substochastic matrix $P \in \mathbb{R}_+^{m \times m}$ is a convex combination of subpermutation matrices.*

*Proof.* We use the proof from [MOA79, Theorem 2.C.2]. We augment the matrix $P$ into a $2m \times 2m$ doubly stochastic matrix as follows:

$$Q = \begin{bmatrix} P & I_m - D_r \\ I_m - D_c & P^T \end{bmatrix}.$$

---

[2]i.e., $f$ is permutation-invariant, and $f(x) = f(|x|)$ for any $x \in \mathbb{R}^n$

where $D_r = \text{diag}(P\mathbf{1})$ and $D_c = \text{diag}(\mathbf{1}^T P)$ are respectively the row and column sums of $P$. The matrix $Q$ is doubly stochastic, and so by the Birkhoff-von Neumann theorem it can be written as a convex combination $Q = \sum_i \alpha_i S_i$ where each $S_i$ is a $2m \times 2m$ permutation matrix. It follows that $P = \sum_i \alpha_i \tilde{S}_i$ where $\tilde{S}_i$ is the $m \times m$ upper-left block of $S_i$. Each $\tilde{S}_i$ is a subpermutation matrix, so this proves the lemma. $\qquad\square$

We will need the following simple additional lemma.

**Lemma 4.** *If $f : \mathbb{R}^m \to \mathbb{R}$ is a convex absolutely symmetric function, and $P \in \mathbb{R}_+^{m \times m}$ is a substochastic matrix, then for any $x \in \mathbb{R}^m$, $f(Px) \leq f(x)$.*

*Proof.* We first show that $f(Sx) \leq f(x)$ if $S$ is a subpermutation matrix. It is not hard to see that if $S$ is a subpermutation matrix then it can be written as $S = (S_1 + S_2)/2$ where $S_1, S_2$ are two *signed permutation matrices*, i.e., permutation matrices where the nonzero entries are $\pm 1$. Since $f$ is convex we get $f(Sx) \leq (f(S_1 x) + f(S_2 x))/2 = f(x)$ where the latter equality follows from $f(S_1 x) = f(S_2 x) = f(x)$ by the fact that $f$ is absolutely symmetric.

If $P$ is substochastic, it is a convex combination of subpermutation matrices, by Lemma 3. Using convexity of $f$ we immediately get that $f(Px) \leq f(x)$. $\qquad\square$

We can now prove (2). The inequality $\leq$ is clear. We focus on $\geq$. Let $U \in O(m)$ and $V \in O(n)$. We know from Lemma 2 that $|\text{diag}(U^T X V)| = P\sigma(U^T X V) = P\sigma(X)$ for some substochastic matrix $P$. Thus we get

$$f(\text{diag}(U^T X V)) = f(|\text{diag}(U^T X V)|) = f(P\sigma(X)) \overset{(*)}{\leq} f(\sigma(X)) = F(X)$$

where inequality (*) follows from Lemma 4. This finishes the proof.

## 2 Proof of Theorems 2 and 3

• We start by proving (i).

Let $X^* = \text{prox}_F(Y)$ be the true proximal point. By definition of $X^*$ we have $0 \in \partial\Phi(X^*; Y)$. To prove the claim it suffices to show that there is a point $V \in \partial\Phi(\hat{X}; Y)$ such that $\|V\|_F \leq \epsilon$. Indeed, with such a $V$, we have using the Cauchy-Schwarz inequality, and the strong convexity of $\Phi$:[3]

$$\|\hat{X} - X^*\|_F^2 \leq \left\langle V - 0, \hat{X} - X^* \right\rangle \leq \|V\|_F \|\hat{X} - X^*\|_F$$

which implies $\|\hat{X} - X^*\|_F \leq \|V\|_F \leq \epsilon$. (This argument is from [Yao16, Prop. 7].)

We now show how to construct such a $V$. Observe that $\partial\Phi(\hat{X}; Y) = \partial F(\hat{X}) + \hat{X} - Y$. Thus we need to exhibit a $G \in \partial F(\hat{X})$ such that $\|G + \hat{X} - Y\|_F \leq \epsilon$. Call $\hat{x} = \text{prox}_f(\hat{y})$. By optimality condition of the prox we have $\hat{y} - \hat{x} \in \partial f(\hat{x})$. It thus follows, by rules for the subgradient (see e.g., [PB14, Section 6.7.2]), that $G = \hat{U}\text{diag}(\hat{y} - \hat{x})\hat{U}^T \in \partial F(\hat{X})$. Now if we call $V = G + \hat{X} - Y$ then $V \in \partial\Phi(\hat{X}; Y)$ and

$$\|V\|_F = \|\hat{U}^T V \hat{U}\|_F = \|\text{diag}(\hat{y} - \hat{x}) + \text{diag}(\hat{x}) - \hat{U}^T Y \hat{U}\|_F$$
$$= \|\text{diag}(\hat{y}) - \hat{U}^T Y \hat{U}\|_F = \|E\|_F \leq \epsilon$$

as desired, where we used the fact that $\hat{U}^T Y \hat{U} = \text{diag}(\hat{y}) + E$ and $\|E\|_F \leq \epsilon$. (Note that we did not use in this proof the fact that $E_{ii} = 0$ for $i = 1, \ldots, n$.)

• We now prove (ii).

Let $\theta : \mathbb{R}^n \to \mathbb{R}$, $\theta(y) = \Phi_f(\text{prox}_f(y); y) := \min_{x \in \mathbb{R}^n} \{f(x) + (1/2)\|x - y\|^2\}$. Note that $\theta$ is a convex function, as it is the partial minimization of the jointly convex function $(x, y) \mapsto f(x) + (1/2)\|x - y\|^2$. Observe that since $f$ is permutation-invariant, $\theta$ is also permutation-invariant.

---

[3]Recall that if $f$ is a 1-strongly convex function then $\|x - y\|^2 \leq \langle g_x - g_y, x - y \rangle$ for any $g_x \in \partial f(x)$ and $g_y \in \partial f(y)$.

On the one hand, we know that

$$\Phi(\operatorname{prox}_F(Y); Y) = \min_{X \in \mathbf{S}^n} \left\{ F(X) + (1/2)\|X - Y\|_F^2 \right\} = \theta(y_1, \ldots, y_n)$$

where $y_1, \ldots, y_n$ are the eigenvalues of $Y$. On the other, we have

$$\Phi(\hat{X}; Y) = F(\hat{X}) + \frac{1}{2}\|\hat{X} - Y\|_F^2 = f(\operatorname{prox}_f(\hat{y})) + \frac{1}{2}\|\operatorname{diag}(\operatorname{prox}_f(\hat{y})) - \operatorname{diag}(\hat{y}) - E\|_F^2$$

where we used the fact that $\|\hat{X} - Y\|_F^2 = \|\hat{U}^T(\hat{X} - Y)\hat{U}\|_F^2$. Now since $E$ is diagonal-free we have $\|\operatorname{diag}(\operatorname{prox}_f(\hat{y})) - \operatorname{diag}(\hat{y}) - E\|_F^2 = \|\operatorname{prox}_f(\hat{y}) - \hat{y}\|^2 + \|E\|_F^2$, and as a result we get:

$$\Phi(\hat{X}; Y) = \theta(\hat{y}) + (1/2)\operatorname{off}(E)^2.$$

Note that $\theta(\hat{y}) = \theta(\operatorname{diag}(\hat{U}^T Y \hat{U})) \leq \theta(y_1, \ldots, y_n)$, where $y_1, \ldots, y_n$ are the eigenvalues of $Y$, by (1) (applied to the function $\theta$) and the assumption that $\theta$ is convex. Since $\theta(y_1, \ldots, y_n) = \Phi(\operatorname{prox}_F(Y); Y)$ it thus follows that

$$\Phi(\hat{X}; Y) \leq \Phi(\operatorname{prox}_F(Y); Y) + \epsilon^2/2$$

as desired.

## 2.1   Proof of Theorem 3

The proof is identical to the symmetric case, so we omit it. The only difference is that in (ii) we need to check that the function

$$\theta(y) := \min_{x \in \mathbb{R}^m} \left\{ f(x) + (1/2)\|x - y\|_2^2 \right\}$$

is absolutely symmetric, assuming that $f$ itself is absolutely symmetric, which is easy to verify.