# OpenReview forum: "Faster proximal algorithms for matrix optimization using Jacobi-based eigenvalue methods"
_NeurIPS.cc/2021/Conference — NeurIPS 2021 Poster_

### Official Review · Reviewer_6wmm · 2021-07-09

**Rating:** 6
**Confidence:** 4

**Summary:**

The paper considers proximal algorithms for convex optimization problems over matrices. When the function to be optimized involves a function of the eigenvalues or singular values of a matrix, then prox methods need to evaluate the proximal operator in each iteration which usually boils down to a full eigenvalue decomposition or SVD. This can be very costly. The paper proposes to use the Jacobi eigenvalue algorithm for computing such decompositions. The Jacobi eigenvalue decomposition can be warm started with the decomposition from the last iteration. On average, a good enough approximation to the eigenvalue decomposition can thus be computed by less than one sweep ($n/2$ parallel Givens rotation) over the matrix. Hence, the evaluation of the proximal operator in ADMM or prox algorithms can be speeded up significantly (by a factor of 5 to 20).

Convergence results are provided and experiments validate the approach and its efficacy.

**Limitations And Societal Impact:**

N/A.

**Main Review:**

The paper is very well written and a pleasure to read. It provides a very nice introduction to the Jacobi eigenvalue algorithm and uses it within the framework of (approximate) proximal methods. I did not check the proofs but the theorems seem correct. The core contribution of the paper is to bring Jacobi's eigenvalue algorithm

Some concerns though:
1. The paper uses the resulting algorithm for solving three problems in the experiments: (i) sparse inverse covariance selection, (ii) semidefinite programming, and (iii) low-rank matrix recovery. All experiments are run on synthetic data. There are better algorithms to solve these problems. For (i), use [1] or [2]. For (ii), make use of interior point methods for problems of this scale since they considerably need fewer iterations. For (iii), use a Frank-Wolfe type algorithm. These are the standard methods to solve each of these problems and they are considerably faster than the ADMM approach presented here. While the presented approach is nice and general, one would not use it in order to solve these problems on real-world machine learning instances. Hence,  I do not know, whether NeurIPS is the right venue for such a paper. Maybe some optimization journal/conference would be better suited.

2. The improvement in speed-up for each eigenvalue / SVD decomposition is provided. It would be nice to also report the total solving time, since this speedup does not translate to an overall speedup, e.g., in the case of the SDP experiment, it seems a factor of two improvement in the end.

3. Comparison to state-of-the-art approaches on real-world problems/data would be nice for an ML conference.

4. It is stated in the introduction that there are first-order methods that do not require solving a full eigenvalue decomposition / SVD but instead only need one eigenvector (Frank-Wolfe type algos for instance). It would be nice to also compare to such algorithms in the experimental section, since they are usually much more efficient and to provide a better global picture.

5. What about interior point algorithms? They need much fewer iterations and since a full eigenvalue decomposition is computed in each iteration, the per-iteration complexity is comparable. Especially the size of the problems (up to 1000 x 1000)in the evaluation suggests using IP methods instead.

So all in all, the paper is really nice, but I am not sure if it is a suitable contribution for the ML community.

References:
[1] Cho-Jui Hsieh, Matyas A. Sustik, Inderjit S. Dhillon, Pradeep Ravikumar. Sparse Inverse Covariance Matrix Estimation Using Quadratic Approximation. In NIPS 2011.
[2] Cho-Jui Hsieh, Matyas A. Sustik, Inderjit S. Dhillon, Pradeep K. Ravikumar, Russell Poldrack. BIG & QUIC: Sparse Inverse Covariance Estimation for a Million Variables. In NIPS 2013.


**Time Spent Reviewing:**

3 hours

---

> ### Author Response · Authors · 2021-08-10
> **Response to review 6wmm**
>
> > The paper is very well written and a pleasure to read. It provides a very nice introduction to the Jacobi eigenvalue algorithm and uses it within the framework of (approximate) proximal methods.
>
> We thank the reviewer for their comments, and we are happy that the reviewer enjoyed reading the paper.
>
> > I did not check the proofs but the theorems seem correct. The core contribution of the paper is to bring Jacobi's eigenvalue algorithm
>
> As with Reviewers BV6g and KM7u, we would like to point out that the core contribution is not only the application of the Jacobi method, but also importantly the theoretical results of Section 4 which provide general bounds concerning the approximate evaluation of the proximal operator of matrix functions.  We believe these theorems are remarkable for a number of reasons:
>
> 1. They bound the distance to the true prox (a quantity that is hard to compute) using a simple quantity which is $\epsilon = off(\hat{U}^T Y \hat{U})$
> 2. The bounds apply to *all* convex spectral functions $F$ and do not depend on any Lipschitz assumption on $F$
> 3. The bounds apply not only to the Jacobi method, but to any other method that can be used to produce approximate diagonalizations $\hat{U}$
> 4. Well-known perturbation bounds for eigenvalue/eigenvector computation (e.g., the Davis-Kahan $\sin \theta$ theorem) suggest that the error would depend on additional factors related to the spectrum of $Y$. Such a dependence would make the theorem very hard to apply. Our bounds are free of such dependence and only depend on $\epsilon$
>
> In fact, the proofs of Theorems 2 and 3 are far from being trivial and rely on a careful application of a variational formula for convex spectral functions. Let us just mention that point (i) of Theorem 2 (the easy part) is itself a significant generalization of the main result of the recent paper [GNR20], which only considered the case where $F$ is the indicator function of the positive semidefinite cone. In fact we expect Theorems 2 and 3 to have applications beyond Jacobi's method, in the context of using inexact proximal methods for matrix optimization.
>
> > Some concerns though:
> >
> > The paper uses the resulting algorithm for solving three problems in the experiments: (i) sparse inverse covariance selection, (ii) semidefinite programming, and (iii) low-rank matrix recovery. All experiments are run on synthetic data. There are better algorithms to solve these problems. For (i), use [1] or [2]. For (ii), make use of interior point methods for problems of this scale since they considerably need fewer iterations. For (iii), use a Frank-Wolfe type algorithm. These are the standard methods to solve each of these problems and they are considerably faster than the ADMM approach presented here. While the presented approach is nice and general, one would not use it in order to solve these problems on real-world machine learning instances. Hence, I do not know, whether NeurIPS is the right venue for such a paper. Maybe some optimization journal/conference would be better suited.
>
> We thank the reviewer for bringing [1] to our attention.
> Concerning semidefinite programs, interior-point methods are much slower for the instances we run. For example, on the instance of size $n=600$, Mosek takes more than 150 seconds whereas the ADMM-based methods considered here take less than 6 seconds.
>
> The examples (i), (ii) and (iii) are presented in the paper to demonstrate the potential of the method on some traditional applications, and we understand that specialized algorithms for each of these problems could be faster. One advantage of our method is its **generality**: it can essentially be applied to any optimization method where an eigenvalue or singular value decomposition is needed. For example it can be applied to more complicated variants of (i) with latent variables, and for Gaussian Conditional Random Fields [1,2] -- for such problems ADMM is a natural algorithm. Also it can be applied to variants of (iii) where a more complicated regularizer involving e.g., Ky-Fan norms are used, see e.g., [3], or when the iterates are not necessarily low-rank.
>
>
> [1] Chandrasekaran et al., Latent Variable Graphical Model Selection via Convex Optimization, Annals of Statistics, 2012
>
> [2] Frot et al., Graphical Model Selection for Gaussian
> Conditional Random Fields in the Presence of
> Latent Variables, Journal of the Americal Statistical Association, 2019
>
> [3] Argyriou et al., On spectral learning, JMLR, 2010
>
> > The improvement in speed-up for each eigenvalue / SVD decomposition is provided. It would be nice to also report the total solving time, since this speedup does not translate to an overall speedup, e.g., in the case of the SDP experiment, it seems a factor of two improvement in the end.
>
> Thank you for this suggestion, we have reported the total running times of the experiments below. In some problems the eigenvalue/SVD decomposition is the only main computational work per iteration (Examples (i) and (iii)) whereas in others (e.g., SDP) a linear system solve is required at each iteration. Depending on the structure and size of this linear system it can be more or less expensive than the time required for the eigenvalue decomposition. We see that in Table 2, using Jacobi's method, the linear solve time becomes the main bottleneck per iteration, whereas with the default methods the projection on the psd cone dominates the time. Some semidefinite programs have linear systems that are easy to solve (e.g., in MaxCut and in some sum-of-squares relaxations) and for these instances we expect the speedup to be larger.
>
> Total running times for examples (in seconds):
>
> (i) Covariance selection
>
> | n | default (CPU) | default (GPU) | Jacobi |
> | --- | --- | --- | --- |
> | 200 | 1.47 |  2.29 |  0.69 |
> | 400 | 4.91 |  5.57 |  1.42 |
> | 600 | 13.33 |  9.19 |  2.77 |
> | 800 | 20.64 | 12.78 |  4.80 |
> | 1000 | 34.62 | 18.45 |  7.27 |
>
> (ii) SDP
>
> | n | default (CPU) | default (GPU) | Jacobi |
> | --- | --- | --- | --- |
> | 200 | 0.43 |  1.37 |  0.35 |
> | 400 | 2.22 |  2.44 |  1.27 |
> | 600 | 5.80 |  5.13 |  3.93 |
> | 800 | 11.14 |  8.13 |  7.45 |
> | 1000 | 30.85 | 19.77 | 12.57 |
>
> (iii) Nuclear-norm regularization (SVD)
>
> | n | default (CPU) | default (GPU) | Jacobi |
> | --- | --- | --- | --- |
> | 200 | 7.16 | 17.37 |  2.15 |
> | 400 | 29.39 | 54.76 |  4.40 |
> | 600 | 71.45 | 108.32 |  8.34 |
> | 800 | 152.16 | 183.04 | 14.42 |
> | 1000 | 306.82 | 335.97 | 21.86 |
>
>
> > Comparison to state-of-the-art approaches on real-world problems/data would be nice for an ML conference. It is stated in the introduction that there are first-order methods that do not require solving a full eigenvalue decomposition / SVD but instead only need one eigenvector (Frank-Wolfe type algos for instance). It would be nice to also compare to such algorithms in the experimental section, since they are usually much more efficient and to provide a better global picture.
>
> Thank you for these suggestions.
> Our main focus in this paper is to present the ideas and develop the theory in the most general way, and demonstrate the potential of the method on traditional applications, comparing it to established baselines. A more specialized and systematic experimental study on real-world data with comparison to dedicated state-of-the-art methods is the subject of future research.
>
> > What about interior point algorithms? They need much fewer iterations and since a full eigenvalue decomposition is computed in each iteration, the per-iteration complexity is comparable. Especially the size of the problems (up to 1000 x 1000)in the evaluation suggests using IP methods instead.
>
> As mentioned above, interior-point methods are not applicable for problems of the scale considered here and are far slower than first-order methods.
>
> > So all in all, the paper is really nice, but I am not sure if it is a suitable contribution for the ML community.
>
> Thank you for your comment.
> We believe that the contributions of this paper are highly relevant to the ML community. Indeed, matrix optimization problems that require eigenvalue or singular value decompositions occur very frequently in ML. Moreover, the proximal operator plays a fundamental role in optimization algorithms for ML, and so there is a need to find efficient algorithms for its computation and approximation. There is already a vast ML literature on proximal algorithms and our paper naturally fits into it. Convex spectral functions and their proximal operator have appeared multiple times in this literature e.g., in matrix decomposition, and multi-task transfer learning. The fact that the proximal operator of such functions can be obtained from an eigenvalue/singular value decomposition (the so-called *transfer principle*) has played an important role in these applications. Theoretically, this paper gives the first, to our knowledge, quantitative version of this principle that bounds the distance to the true prox in terms of the quality of the approximate decomposition. Practically, we demonstrate the strong potential of the Jacobi method to speed up the computation of the proximal operator at every iteration.

---

### Official Review · Reviewer_gv2k · 2021-07-15

**Rating:** 6
**Confidence:** 4

**Summary:**

This paper proposes a new method to accelerate proximal-mapping based algorithms  where the proximal mapping usually needs computing the eigen/singular value decomposition.   The authors discusses how to use the Jacobi method to get the eigen/singular value decomposition approximately and parallelly,  and the convergence can be guaranteed by easily controlling the norm of the off diagonal elements.
Numerical experiments show the significant improvement over the state-of-art algorithms.

**Limitations And Societal Impact:**

Not apply.

**Main Review:**

This paper is well-organized and well-written in general.  I have the following main comments:
1. The property of the spectral function in Theorem 1 will play a key role in proving Theorem 2. I suggest clarifying it in section  2.

2. The assumptions on $F(X)$ should be put in  the assumption environment so that the statement of Theorem 2 becomes more clear.

3. It says 'One consequence of this is that the Jacobi method can have faster convergence if off(A) is small, whereas
  this is not the case of other algorithms that start by a reduction into tridiagonal form.' I'm not sure about this. Could you give more details in why the warm-start does not work in other eigenvalue algorithms? For example, can this be verified by experiments? The current implementation in NumPy solves the eigen decomposition exactly.

Minor comment:
The stopping criterion of in Section 5.1 and 5.3 should be given.




**Time Spent Reviewing:**

5

---

> ### Author Response · Authors · 2021-08-10
> **Response to review gv2k**
>
>
> We thank the reviewer for their comments and suggestions.
>
> > The property of the spectral function in Theorem 1 will play a key role in proving Theorem 2. I suggest clarifying it in section 2.
> > The assumptions on  should be put in the assumption environment so that the statement of Theorem 2 becomes more clear.
>
> Thank you for these suggestions. We will take them into account when preparing the final version.
>
> > It says 'One consequence of this is that the Jacobi method can have faster convergence if off(A) is small, whereas this is not the case of other algorithms that start by a reduction into tridiagonal form.' I'm not sure about this. Could you give more details in why the warm-start does not work in other eigenvalue algorithms? For example, can this be verified by experiments? The current implementation in NumPy solves the eigen decomposition exactly.
>
> Current state-of-the-art methods such as QR and divide-and-conquer for computing eigenvalue and singular value decompositions start by putting the matrix into tridiagonal form via a sequence of Householder reflections (a matrix is tridiagonal when the only nonzero entries $A_{i,j}$ are those for which $|i-j| \leq 1$), see e.g., the book Matrix Computations by Golub and van Loan. As a result, such methods cannot exploit a situation where the input matrix is *almost* diagonal. Quoting from the book cited above (page 481) ``*Moreover, if an approximate eigenvector matrix $V$ is known, then $V^T A V$ is almost diagonal, a situation that Jacobi can exploit but not QR*''.
>
> One can indeed verify this fact experimentally. The table below shows the time taken for the Jacobi method to compute the singular value decomposition of a $n\times n$ matrix $A$ of the form $A=A_0 + E$ where $A_0$ is diagonal, and $E$ satisfies $\|E\|_F \leq 10^{-3}$, without any bound on the number of sweeps.
>
> | n | default (NumPy) | jacobi |
> | --- | --- | --- |
> | 1024 | 0.3s | 0.002s |
> | 2048 | 3.5s | 0.07s |
> | 4096 | 25s | 0.5s |
>
> > Minor comment: The stopping criterion of in Section 5.1 and 5.3 should be given.
>
> Thank you for this comment, we will take it into account when preparing the final version. In 5.1 we used the same stopping criterion as the Matlab implementation of [BPC11]. For 5.3 there is no stopping criterion, as we ran the algorithm for a fixed number of iterations.

---

### Official Review · Reviewer_KM7u · 2021-07-16

**Rating:** 7
**Confidence:** 4

**Summary:**

This paper proposes using a classical but often unused method for computing the eigenvalue decomposition of a symmetric matrix: the Jacobi method. The method simply iterates by zeroing out off-diagonal entries using Givens rotations one at a time, until the matrix converges to a diagonal matrix. It is further noted that it is possible to parallelize the removal of ~n/2 entries at a time if disjoint indices are picked (for a matrix of size n-by-n).

Proximal methods are a popular choice for solving convex optimization problems over matrices. A commonly required oracle for such problems is the computation of the (full) eigenvalue decomposition of the current iterate, e.g., to compute the projection of the iterate onto the positive semidefinite cone. It is observed that as the iterations progress, the distance between iterates decreases and so the eigenvalue decomposition of consecutive iterates is not perturbed too much. Therefore, it is expected that few sweeps of the Jacobi method are required to compute the (approximate) eigenvalue decomposition of the next iterate.

The paper further provides bounds on the error of the prox operator as a function of the Frobenius norm of the matrix of off-diagonal entries of the iterate of the Jacobi method. This allows one to establish termination criteria in order to run the Jacobi method to obtain an approximate eigenvalue decomposition, while guaranteeing convergence of the (approximate) proximal algorithm.

The authors provide some numerical experiments on several matrix optimization problems such as solving sparse inverse covariance selection, SDPs, nuclear norm minimization, for small to moderately sized problems. Their experiments show that using the Jacobi algorithm for a small number of sweeps obtains sufficient accuracy to proceed to the next iteration of the proximal algorithm, and therefore reduces the overall wall-clock time to convergence, compared to using an eigenvalue/singular value decomposition implemented on CPUs or GPUs.

**Ethical Concerns:**

N/A.

**Limitations And Societal Impact:**

As discussed in the previous section, my main concern is over the numerical experiments, and seeing this applied on truly large scale problems, rather than those that can be solved with conventional approaches quickly already.

I do not see any issues regarding this work and potential negative impacts on society.

**Main Review:**

Originality:
I've not seen the Jacobi algorithm applied in this way, so as far as I can tell, the result is original. Though most results in this paper are not particularly new, there is merit in combining the ideas of using a few Jacobi sweeps (with guaranteed and easy to compute convergence criteria).

Quality:
Overall, the results of the paper appear sound, though I have some comments about certain results and experiments.
  - Beginning of section 5: I do not see the significance of comparing a single sweep of the Jacobi algorithm against the computation of the full eigenvalue/singular value decomposition. A single sweep will not compute anything meaningful and obviously does less work than any full decomposition, so Table 1 is not helpful and in general misleading
  - Numerical experiments in general: these problems are quite small by today's standards, and the iteration time per problem is quite fast (on the order of ms) for all methods involved. Especially given the message of the paper, which focuses on exploiting parallelism for large scale problems, it is necessary to increase the size of the problems in the experiments section.
  - Remark 1: what floating point precision was used in your experiments? It is clear that continuously applying givens rotations would result in loss of orthogonality over time, but this can be addressed by re-orthogonalizing U periodically (which s equivalent to 1 sweep of Jacobi; in fact, it should be faster since it can be implemented using BLAS 3 operations on the GPU).

Some other comments:
  - It would be helpful to include convergence results for the cyclic Jacobi methods. The authors provide convergence results for the classical method that picks the largest off-diagonal entry, but this algorithm isn't used in this paper.
  - Out of curiosity, are there convergence results for a randomized Jacobi algorithm.
  - For each experiment, what was the fraction of time spent running the Jacobi sweep vs. computing the Frobenius norm of the error? In fact, how often was the Frobenius error computed to determine whether sufficient accuracy was reached?

Clarity:
The paper is overall well-written, and I think communicates its ideas well.

Significance:
If this paper can be shown to work well on truly large scale problems, then it's significance would be improved. However, with the current experiments, it's difficult to say that it's a truly significant result.

**Time Spent Reviewing:**

4

---

> ### Author Response · Authors · 2021-08-10
> **Response to review KM7u**
>
> We would like to thank the reviewer for their careful review of the manuscript.
>
> > Originality: I've not seen the Jacobi algorithm applied in this way, so as far as I can tell, the result is original. Though most results in this paper are not particularly new, there is merit in combining the ideas of using a few Jacobi sweeps (with guaranteed and easy to compute convergence criteria).
>
> We would like to point out that the theoretical results of Section 4 are completely new to the best of our knowledge. In fact, we see Section 4 as a core contribution of the paper. As mentioned with Reviewers BV6g and 6wmm, we believe the theorems proved in this section are remarkable for a number of reasons:
>
> 1. They bound the distance to the true prox (a quantity that is hard to compute) using a simple quantity which is $\epsilon = off(\hat{U}^T Y \hat{U})$
> 2. The bounds apply to *all* convex spectral functions $F$ and do not depend on any Lipschitz assumption on $F$
> 3. The bounds apply not only to the Jacobi method, but to any other method that can be used to produce approximate diagonalizations $\hat{U}$
> 4. Well-known perturbation bounds for eigenvalue/eigenvector computation (e.g., the Davis-Kahan $\sin \theta$ theorem) suggest that the error would depend on additional factors related to the spectrum of $Y$. Such a dependence would make the theorem very hard to apply. Our bounds are free of such dependence and only depend on $\epsilon$
>
> In fact, the proofs of Theorems 2 and 3 are far from being trivial and rely on a careful application of a variational formula for convex spectral functions. Let us just mention that point (i) of Theorem 2 (the easy part) is itself a significant generalization of the main result of the recent paper [GNR20], which only considered the case where $F$ is the indicator function of the positive semidefinite cone. In fact we expect Theorems 2 and 3 to have applications beyond Jacobi's method, in the context of using inexact proximal methods for matrix optimization.
>
> > Quality: Overall, the results of the paper appear sound, though I have some comments about certain results and experiments. Beginning of section 5: I do not see the significance of comparing a single sweep of the Jacobi algorithm against the computation of the full eigenvalue/singular value decomposition. A single sweep will not compute anything meaningful and obviously does less work than any full decomposition, so Table 1 is not helpful and in general misleading
>
> The reason we compare a single sweep of the Jacobi algorithm against the default method is that, as the experiments suggest (see Tables 1,2,3), the number of Jacobi sweeps that are needed at each iteration of the optimization algorithm is approximately equal to 1. This is because, at each iteration, the matrix we need to decompose is ``almost-diagonal'' (see Section 3.2). On such almost-diagonal matrices, the Jacobi method is able to compute a full and accurate diagonalization much faster than other methods.
>
> > Numerical experiments in general: these problems are quite small by today's standards, and the iteration time per problem is quite fast (on the order of ms) for all methods involved. Especially given the message of the paper, which focuses on exploiting parallelism for large scale problems, it is necessary to increase the size of the problems in the experiments section.
>
> Following the reviewer's suggestion we have done additional experiments on larger problems involving the singular value decomposition (Section 5.3). We considered a problem of size $n=5000$ where the default baseline takes ~25s per iteration (fast proximal gradient method with full SVD done on the GPU). After 2 hours of running time, the objective value attained is the same as the one attained by the proposed Jacobi method in 8 minutes, which corresponds to a speed-up of ~14x. We will include a more detailed plot of the evolution of both methods vs. time in the final version of the paper.
>
>
> For problems involving symmetric matrices, we have noticed that the corresponding Jacobi function ```syevj``` (provided by Nvidia's cusolver) suffers a significant degradation in performance for $n$ larger than 2000 which makes it not competitive with the default approaches (non-Jacobi). It is not clear to us what is the cause of this degradation and we are planning to investigate this in the future. We believe the implementation may have been specifically tuned/designed for moderate size matrices, and we are considering writing our own implementation of the Jacobi method.
>
> > Remark 1: what floating point precision was used in your experiments? It is clear that continuously applying givens rotations would result in loss of orthogonality over time, but this can be addressed by re-orthogonalizing U periodically (which s equivalent to 1 sweep of Jacobi; in fact, it should be faster since it can be implemented using BLAS 3 operations on the GPU).
>
> Thank you for this suggestion. We used float32 for the Jacobi GPU computations. Indeed, we observed that periodically re-orthogonalizing $U$ using a QR factorization is helpful in maintaining orthogonality of $U$ throughout iterations. We used this approach in the example of size $n=5000$ mentioned above.
>
> > It would be helpful to include convergence results for the cyclic Jacobi methods. The authors provide convergence results for the classical method that picks the largest off-diagonal entry, but this algorithm isn't used in this paper.
>
> Thank you for this suggestion, we will include such results in the final version of the paper. Note that we cite in Section 3.1 references that analyze the cyclic method [FH60,Wil62,VK66b], however we did not formally state the convergence results.
>
> > Out of curiosity, are there convergence results for a randomized Jacobi algorithm.
>
> We are not aware of any result concerning the randomized Jacobi algorithm.
>
> > For each experiment, what was the fraction of time spent running the Jacobi sweep vs. computing the Frobenius norm of the error? In fact, how often was the Frobenius error computed to determine whether sufficient accuracy was reached?
>
> The Frobenius norm was computed at every iteration. The time taken for this is negligible vs. the time taken to do a single Jacobi sweep.
>
> > Significance: If this paper can be shown to work well on truly large scale problems, then it's significance would be improved. However, with the current experiments, it's difficult to say that it's a truly significant result.
>
> We hope that the results shown above for SVD demonstrate that the proposed approach does scale to large problems.
> As mentioned above, we also think that the significance of this paper is in the theoretical part.

---

### Official Review · Reviewer_BV6g · 2021-07-16

**Rating:** 6
**Confidence:** 4

**Summary:**

This work proposes to apply a Jacobi-based approximate eigenvalue decomposition method in proximal slitting algorithms for minimization of convex spectral function with matrix variable. The adopted Jacobi-based method is used to compute proximal operators in each iteration, and by taking the previous iterate as the initial point, the Jacobi-based method appears to reach required accuracy within fewer sweeps. The method can be parallelized, and its GPU implementation shows improvement in efficiency compared with standard eigenvalue decomposition functions in LAPACK and CUSOLVER in numerical experiments.

**Limitations And Societal Impact:**

It would be better to provide certain bound on the complexity of the method and to compare it with existing randomized approximate eigenvalue decomposition methods, in order to better demonstrate its advantage and limitation.

**Main Review:**

The Jacobi-based approximate eigenvalue decomposition method is not novel, and its cyclic and parallelized implementation is also not new. These are introduced in section 3.1. There is certain novelty in applying the method to compute proximal operators in proximal algorithms, which is introduced in section 3.2. In order to better clarify the contribution, it is suggested that section 3.1 and 3.2 should be separated into two different sections, since the former is a review of existing work, and the latter is the main contribution.

The significance and novelty can be improved by bringing more theoretical insights. For example, is the number of sweeps bounded by the increment between the previous and current iterates? How many sweeps would be required for a certain class of proximal algorithms such as ADMM?

Randomized approximate eigenvalue decomposition (or singular value decomposition) has been an active research topic. How is the adopted Jacobi-based method compared with these randomized methods? It is suggested that such discussion should be added in the related work, and in the experiments the proposed approach should be compared with these methods.

From the experimental results in Table 1, one sweep of the Jacobi method is about 10 times faster when n is moderate (256 or 512), but the improvement is only about 3 times when n is large (2048). Does it indicate that the advantage would become less and less when n scales up?

In the experiments, there are some parameters in the accuracy $\epsilon_k$, namely, the scale taking values 0.1 and 1 and the power taking value 2.5. Are these parameters carefully tuned? Are their choices crucial to the performance?

**Time Spent Reviewing:**

8 hours

---

> ### Author Response · Authors · 2021-08-10
> **Response to review BV6g**
>
> We would like to thank the reviewer for their careful review of the manuscript.
>
> > In order to better clarify the contribution, it is suggested that section 3.1 and 3.2 should be separated into two different sections, since the former is a review of existing work, and the latter is the main contribution.
>
> Thank you for this suggestion, which we will take into account in the final version of the paper.
> Concerning the main contribution of the paper, we would like to emphasize that the main contribution is not only Section 3.2 (the application of Jacobi method), but importantly the new theorems Theorems 2 and 3 in Section 4. We believe these theorems are remarkable in a number of ways:
> 1. They bound the distance to the true prox (a quantity that is hard to compute) using a simple quantity which is $\epsilon = off(\hat{U}^T Y \hat{U})$
> 2. The bounds apply to *all* convex spectral functions $F$ and do not depend on any Lipschitz assumption on $F$
> 3. The bounds apply not only to the Jacobi method, but to any other method that can be used to produce approximate diagonalizations $\hat{U}$
> 4. Well-known perturbation bounds for eigenvalue/eigenvector computation (e.g., the Davis-Kahan $\sin \theta$ theorem) suggest that the error would depend on additional factors related to the spectrum of $Y$. Such a dependence would make the theorem very hard to apply. Our bounds are free of such dependence and only depend on $\epsilon$
>
> In fact, the proofs of Theorems 2 and 3 are far from being trivial and rely on a careful application of a variational formula for convex spectral functions. Let us just mention that point (i) of Theorem 2 (the easy part) is itself a significant generalization of the main result of the recent paper [GNR20], which only considered the case where $F$ is the indicator function of the positive semidefinite cone. In fact we expect Theorems 2 and 3 to have applications beyond Jacobi's method, in the context of using inexact proximal methods for matrix optimization.
>
> > The significance and novelty can be improved by bringing more theoretical insights. For example, is the number of sweeps bounded by the increment between the previous and current iterates? How many sweeps would be required for a certain class of proximal algorithms such as ADMM?
>
> Section 4 is devoted to a theoretical study of approximate computation of proximal operators for spectral functions, and provides termination criteria that guarantee convergence of the overall optimization problem.
>
> The questions raised by the reviewer are very interesting. Existing literature on the Jacobi method show that the method is locally quadratic convergent, i.e., $off(A^+) \leq c \cdot off(A)^2$, where $A^+$ is the matrix obtained after one sweep. One can, *a priori*, use such results to obtain a quantitative estimate of the number of sweeps at each iteration, however such results would depend on the constant $c$ that depends on spectral properties of the iterates $A_k$ and is hard to estimate. The question of finding explicit bounds is a very interesting one nevertheless, and we leave it for future work.
>
> > Randomized approximate eigenvalue decomposition (or singular value decomposition) has been an active research topic. How is the adopted Jacobi-based method compared with these randomized methods? It is suggested that such discussion should be added in the related work, and in the experiments the proposed approach should be compared with these methods.
>
> Thanks for the suggestion, we will add a reference to such methods in the related work.
> We note however that such methods are only applicable when the iterates are known to be low-rank, and a bound on their rank is known. In contrast, our methods apply generally to problems where the matrix iterates do not necessarily have low rank (such as e.g., in covariance estimation, or semidefinite programming).
>
> > From the experimental results in Table 1, one sweep of the Jacobi method is about 10 times faster when n is moderate (256 or 512), but the improvement is only about 3 times when n is large (2048). Does it indicate that the advantage would become less and less when n scales up?
>
> The implementation we have used for the Jacobi method (provided by Nvidia's cusolver package) for symmetric matrices seems indeed to attain its best performance for $n \sim 500$ and we have observed that, surprisingly, there is a performance degradation for $n \geq 2000$. It is unclear to us what is the reason of this performance degradation and this is something we are currently investigating.
>
> Let us mention that this performance degradation does not happen for the singular value decomposition. The table below extends Table 1 to larger matrix sizes for the SVD, and we see that 1 sweep of Jacobi method is still significantly faster than the default algorithms for large matrix sizes.
>
> | n | default (CPU) | default (GPU) | jacobi-1sw (GPU) |
> | --- | --- | --- | --- |
> | 4096 | 25s | 20s | 0.5s |
> | 8192 | 178s | 144s | 4s |
>
> > In the experiments, there are some parameters in the accuracy , namely, the scale taking values 0.1 and 1 and the power taking value 2.5. Are these parameters carefully tuned? Are their choices crucial to the performance?
>
> The rate of decay of $\epsilon_k$ follows from the theoretical results of Section 4: the choice $\epsilon_k \propto k^{-2.5}$ guarantees a rate of convergence of $1/k^2$ for the fast proximal gradient method. For the ADMM, summability of $(\epsilon_k)$ is enough, however we saw that choosing $k^{-2.5}$ does not significantly increase the number of sweeps per iteration.
>
> The choice of the constant typically depends on the algorithm (ADMM, fast proximal gradient, ...) and the scaling of the problem. Choosing a large constant can reduce the cost per iteration (number of sweeps) but can increase the total number of iterations of the optimization. For example on the covariance selection problem of size $n=1000$ changing the constant from 0.1 to 1.0 reduces the sweeps/iter from 0.94 to 0.74 but increases the total number of iterations from 230 to 243. The effect on the total running time is negligible in this case.
>
> Finally we mention that another possibility is to keep a constant number of sweeps at each iteration, e.g., 1. We found that this option also performs  well in practice.

---

> > ### Comment · Reviewer_BV6g · 2021-08-31
> > **Re: Response to review BV6g**
> >
> > Thank the authors for answering my comments. After reading the response, I find that the theoretical contribution in Section 4 is more interesting than I thought in the first review, so I will increase my rating to 6.

---

### Decision · Program_Chairs · 2021-09-27

**Decision:**

Accept (Poster)

**Comment:**

The submission studies the application of the Jacobi method to compute approximate eigendecompositions or SVD in the context of solving composite optimization problems over matrices, where the objective functions are spectral. The main theoretical contribution of the paper shows how the error in the approximate decomposition (in terms of the $\ell_2$-norm of the off-diagonal entries) propagates into the analysis of proximal gradient descent and proximal accelerated gradient descent. The AC agrees with the reviewers that this is an interesting result, albeit fairly simple given previous work of inexact oracles for proximal methods. Moreover, the reviewers and the AC all find that the paper is well-written, making the technical contribution easy to grasp.

The main practical contribution is an experimental evaluation of the resulting Jacobi-based method against the  analogue QR-based methods, which generally shows the superiority of the Jacobi method. This is partly explained by the fact that the Jacobi method can effectively use previous decompositions as warm starts for the next. The AC agrees with reviewers BV6g and KM7u that the significance of such experiments would be increased by including larger matrices and real-world datasets.

In conclusion, the AC believes that the interesting and well-explained theoretical contribution, together with the positive results on the experimental part, outweigh the lack of larger experiments and recommends that the paper be accepted.